# Neutralising Effects of Different Antibodies on *Clostridioides difficile* Toxins TcdA and TcdB in a Translational Approach

**DOI:** 10.3390/ijms24043867

**Published:** 2023-02-15

**Authors:** Georg Csukovich, Nina Kramer, Barbara Pratscher, Ivana Gotic, Patricia Freund, Rainer Hahn, Gottfried Himmler, Sabine Brandt, Iwan Anton Burgener

**Affiliations:** 1Small Animal Internal Medicine, Department for Companion Animals and Horses, Vetmeduni, 1210 Vienna, Austria; 2The Antibody Lab GmbH, 1210 Vienna, Austria; 3Department for Biotechnology, University of Natural Resources and Life Sciences Vienna, 1190 Vienna, Austria; 4Research Group Oncology (RGO), Clinical Unit of Equine Surgery, Department for Companion Animals and Horses, Vetmeduni, 1210 Vienna, Austria

**Keywords:** organoids, *Clostridioides difficile*, sIgA, neutralisation, toxin

## Abstract

Given the high prevalence of intestinal disease in humans and animals, there is a strong need for clinically relevant models recapitulating gastrointestinal systems, ideally replacing in vivo models in accordance with the principles of the 3R. We established a canine organoid system and analysed the neutralising effects of recombinant versus natural antibodies on *Clostridioides difficile* toxins A and B in this in vitro system. Sulforhodamine B cytotoxicity assays in 2D and FITC-dextran barrier integrity assays on basal-out and apical-out organoids revealed that recombinant, but not natural antibodies, effectively neutralised *C. difficile* toxins. Our findings emphasise that canine intestinal organoids can be used to test different components and suggest that they can be further refined to also mirror complex interactions between the intestinal epithelium and other cells.

## 1. Introduction

Functional gastrointestinal (GI) disorders constitute a major, potentially lethal, health problem in humans and animals. It is accepted today that the intestinal microbiome and secretory immune system have a crucial role in maintaining homeostasis between various genera of microorganisms, thereby protecting the GI tract from the detrimental effects of infectious agents [1,2,3]. Gastrointestinal disorders, including GI cancer, inflammatory bowel disease (IBD) and infectious GI diseases, have a major negative impact on human health and impose a high financial burden on healthcare systems. Infectious diarrheal disease is the second leading cause of death among young children, and GI cancer is responsible for about three million deaths per year worldwide [4].

Potentially lethal GI diseases also affect farm and companion animals, with enterotoxigenic bacteria and enteropathogenic viruses being frequently involved in disease onset and progression [5,6,7]. One important bacterium in this regard, *Clostridioides difficile* (*C. difficile*), formerly known as *Clostridium difficile*, is an anaerobic, Gram-positive bacterium with zoonotic potential, which is widespread all over the world. Faeces of infected animals contain *C. difficile* spores that contaminate soil and water, leading to the propagation of the infection to other animals and humans. Similarly, spores originating from affected humans end up in wastewater, which makes *C. difficile* a global issue for humans, animals and the environment in the context of “One Health” [8]. The importance of animals as symptomatic and asymptomatic carriers of *C. difficile* has been well documented in recent times [9,10]. Although isolates of *C. difficile* commonly cluster based on host species, co-clustering of isolates from cows and dogs with those of paediatric human patients recently emphasised the possibility of interspecies transmission via direct contact or a contaminated environment [11].

In humans, it is estimated that breastfeeding would reduce the risk of dying from diarrhoea by twenty times in new-borns, as breast milk represents a rich reservoir of antimicrobial immunoglobulins (Igs) [12]. As with human neonates [13], the immune status of new-born dogs depends greatly on colostrum (and milk) ingestion, since canine neonates are usually agammaglobulinemic [14]. After the closure of the intestinal barrier, 90 to 95% of circulating Igs originate from the colostrum. Inadequate colostrum intake and suckling lead to a deficit in the transfer of passive immunity that is associated with considerably higher mortality and morbidity rates [15].

Secretory IgA (sIgA) is the predominant Ig protecting mucosal surfaces [16]. Polymeric sIgAs consist of two to four IgA monomers, which are linked by the joining (J) chain and a heavily glycosylated secretory component (SC) [17]. The SC protects the sIgA complex from proteolytic cleavage and binds microbial antigens and certain receptors via specific glycan structures [18]. sIgA can be purified in sufficient amounts from milk or whey of animals such as goats and cows and could potentially be used for the prevention of certain GI disorders [19,20,21]. In particular, glycosylation patterns of caprine and human sIgA show a high similarity [22]. Human colostrum-derived sIgA molecules were previously shown to inhibit the binding of *C. difficile* toxin A (TcdA) to intestinal membranes [18] and partly showed neutralising activity against toxin A and toxin B (TcdB) [23]. Goat milk-derived sIgA isolates from different breeds and lactation periods were investigated for their neutralising potential to TcdA as well as lipopolysaccharide from *Escherichia coli* (*E. coli)* and *Salmonella typhimurium*, the heat-sensitive toxin from *E. coli*, and proteoglycan from *Staphylococcus aureus*. Results obtained by immunoassays reflect a broad spectrum of toxin binding capacity of caprine sIgA with sample-specific variations [24].

Given the high prevalence of intestinal disease in humans and animals, there is a strong need for clinically relevant models of the GI system, since preclinical GI translational research still relies entirely on animals, more specifically, on rodent models that are of limited physiological relevance. As reviewed by Jiminez et al. (2015) intestinal inflammation in rodents differs from naturally occurring diseases in humans, so rodent models are not best suited for the study of this type of disorder [25]. Dogs, however, spontaneously develop chronic enteropathies resembling those in humans, and several subtypes with yet unknown aetiology can be differentiated [26].

Intestinal organoids allow the study of interactions between the gut microbiota and gut epithelium. Hence, exploring ways to mitigate the disease-provoking effects of pathogens and/or boost the health-promoting effects of natural commensals is possible using intestinal organoids, as we have recently outlined [27]. We have previously established and characterised canine intestinal organoid models, such as jejunal and colonic organoids, which can function as epithelial models that faithfully mimic respective intestinal sections [28]. These models have the potential to help close the gap between in vitro screening and in vivo assessment of possible therapeutic drugs, in adherence to the principle of the 3R, i.e., Replacement, Reduction and Refinement, of animal experimentation.

Herein, we report on the use of jejunal and colonic organoids to analyse the neutralising effects of recombinant versus natural antibodies on TcdA and TcdB. Recombinant antibodies comprised the IgG antibody bezlotoxumab, which is in clinical use for the treatment of *C. difficile* infections [29,30], and a recombinant sIgA established by us [31]. Natural antibodies consisted of sIgA purified from pooled goat whey. Analyses of the respective binding affinities of these antibodies to both toxins were carried out by enzyme-linked immunosorbent assay (ELISA). Subsequently, sulforhodamine B (SRB) cytotoxicity and FITC-dextran barrier integrity assays were used to establish the antibodies’ respective toxin neutralisation profiles in the organoid model. We show that recombinant sIgA proved equally effective in neutralising the cytotoxic effect of TcdB on canine intestinal organoids and organoid-derived monolayers (ODM), whilst goat sIgA failed to sufficiently neutralise TcdB in these in vitro systems. This highlights the applicability of canine intestinal organoids for drug testing and the development of treatments.

## 2. Results

### 2.1. Recombinant and Natural sIgA Antibodies Were Successfully Purified

Recombinant anti-TcdB sIgA antibodies used in this study were obtained by co-incubating cell culture supernatants from mIgA2-expressing and hSC-expressing recombinant CHO-K1 cell lines described previously [31]. Free dimeric IgA molecules that did not complex with the hSC were discarded by anion exchange chromatography (AIEx) (Figure 1A). Since this purification step failed to also remove free hSC and monomeric IgA (mIgA), size exclusion chromatography was carried out, resulting in pure sIgA (Figure 1B). sIgA from caprine whey was purified by ultra/diafiltration and size exclusion chromatography (Appendix A). Most fractions obtained comprised both sIgM and sIgA, with higher sIgA content being noted for later fractions (Figure 1C).

### 2.2. Recombinant and Natural Antibodies Mainly Bind to TcdB

The binding potential of bezlotoxumab, recombinant monoclonal sIgA and three sIgA fractions from caprine whey with the highest sIgA content according to SDS-PAGE analyses against TcdA and TcdB was analysed via ELISA. All antibodies showed very low binding potential against TcdA compared to TcdB. However, bezlotoxumab and recombinant sIgA were highly specific against TcdB. Caprine whey fractions showed steadily increasing values with increasing concentrations but remained lower in their highest concentration (1600 µg/mL) than bezlotoxumab and recombinant sIgA at 50 µg/mL. Of all whey isolates, fraction 5B3 scored the best values. Therefore, it was used for all consecutive experiments (Figure 2).

### 2.3. Recombinant Antibodies Exhibited Superior TcdB Neutralisation Potential Compared to Natural sIgA Molecules

The cytotoxic effect of TcdA and TcdB on canine intestinal organoid-derived monolayers was addressed by sulforhodamine B assays. Both small and large intestinal ODMs exhibited significantly decreased viability when treated with either toxin alone or a combination of the two. The attempt to neutralise TcdA failed with all three tested antibodies, i.e., viability was still significantly decreased upon TcdA treatment compared to untreated controls (Figure 3A,B). In contrast, pre-incubation of TcdB with 50 µg/mL of bezlotoxumab or recombinant sIgA resulted in toxin neutralisation, with canine epithelial cells remaining unaffected by TcdB. Pre-incubation of TcdB with caprine sIgA did not increase the viability of TcdB-treated ODMs (Figure 3C,D). All ODMs treated with a combination of TcdA and TcdB showed significantly reduced viability compared to untreated controls. Nonetheless, bezlotoxumab and recombinant sIgA were able to mitigate the toxins’ negative impact on cell viability to some extent (Figure 3E,F).

### 2.4. Successful Establishment of Apical-Out and Floating Basal-Out Organoids

In order to make the apical cell surface accessible to assess the effect of bacterial toxins, floating basal-out and apical-out organoids were established. Basal-out and apical-out organoids were of substantially different appearances, as documented by bright field imaging. Whilst basal-out organoids exhibited a large lumen, apical-out organoids appeared more compact and dense and remained relatively small after polarity reversal (Figure 4A). DAPI/Phalloidin co-staining of DNA and F-actin demonstrated the localisation of apical microvilli oriented towards the lumen in basal-out organoids [32]. In contrast, microvilli, i.e., the apical cell surface, were clearly presented away from the organoid lumen, therefore demonstrating successful polarity reversal (Figure 4B,C). Virtually all organoids turned apical-out three days after initiation of polarity reversal.

### 2.5. Considerably Different Effects of TcdA and TcdB on Basal-Out and Apical-Out Organoids

Since TcdA and TcdB were shown to have a negative impact on cell survival in ODM culture, small and large intestinal organoids were subjected to a FITC-Dextran barrier integrity assay (Figure 5). Toxicity likely depends on the presence of specific receptors on certain cell surface domains. In dogs, these receptors remain to be identified. Hence, we opted for using basal-out and apical-out organoids to study the cytotoxic effects of TcdA and TcdB on either side of the epithelial cells. As shown in Figure 5, TcdA negatively affected cell–cell contacts in all tested organoids, i.e., basal-out and apical-out small and large intestinal organoids (Figure 5B–E). In contrast, TcdB had a negative impact only on barrier integrity in basal-out organoids of the small intestine (Figure 5B). Importantly, this cytotoxic effect of TcdB could be prevented by the pre-incubation of the toxin with bezlotoxumab or recombinant sIgA (Figure 5B,D). This TcdB neutralising effect was also evident in combination with TcdA in small and large intestinal basal-out organoids. Representative confocal microscopy images illustrating these observations are provided in Appendix A.

## 3. Discussion

*Clostridioides difficile* is one of the most important pathogens associated with hospital-associated infections leading to diarrhoea and pseudomembranous colitis. As extensively reviewed by Lim, Knight and Riley, 2020, *C. difficile* poses a major threat in the context of One Health. *Clostridioides difficile* spores can resist severe environmental conditions and be reactivated to germinate in various different hosts, making pet animals a potential source of human infections and vice versa, although exact mechanisms of transmission remain to be clarified. Spores can persist in environmental settings (water, soil, floors, foods) and then infect human beings or animals who have previously undergone antibiotic therapy or reside in different hosts in a latent form [33,34]. Two studies have previously shed light on the importance of asymptomatic carriers of *C. difficile* as an infection source. According to these studies, 10–15% of humans are colonised by *C. difficile* strains, with more than 80% of them constituting toxigenic strains [35,36]. In dogs, apparently, 0–6% of healthy individuals carry *C. difficile* strains, especially those also commonly found in humans [34,37]. However, this percentage might be underestimated, as healthy pets are not as accessible for infection studies as healthy humans.

Given the pathogenic impact of *C. difficile* on human and animal gastrointestinal health, we analysed the potential of two recombinant versus a natural antibodies to bind and neutralise relevant *C. difficile* toxins, i.e., TcdA and TcdB, in canine organoids and ODMs. The studied antibodies comprised the anti-TcdB IgG antibody bezlotoxumab, a recombinant anti-TcdB sIgA antibody established by us, and natural sIgA antibodies isolated from goat whey [31].

In the next step, we addressed the impact of pre-incubation of TcdA, TcdB, or both toxins with the antibody on their individual and cumulative cytotoxic effect on ODMs by employing a sulforhodamine B assay for a more definite outcome from only healthy, living cells attached to the surface and excluded dead cells from the analysis. In agreement with conformed binding affinities to TcdB, bezlotoxumab and recombinant anti-TcdB sIgA effectively neutralised TcdB, thus abrogating its cytotoxic potential, whilst caprine sIgA had no neutralising effect. Similarly, the low binding affinity of recombinant antibodies to TcdA was translated into low neutralisation of the toxin and reduced viability of exposed ODMs. Although binding of caprine sIgA to TcdA could be demonstrated, albeit at a low level, the natural antibody failed to neutralise the toxin and protect ODMs from its cytotoxic effect. This could be due to insufficient amounts of specific sIgA molecules in the whey. However, applying a 20-fold amount of caprine sIgA to the cells was insufficient to neutralise TcdA or TcdB, too, (not shown) despite cow milk sIgA having been already shown to effectively neutralise TcdA [38]. However, this positive effect may depend heavily on the species from which the milk is derived [12]. The combination of both toxins had a more detrimental effect than individual toxins, as reflected by a severe decrease in cell viability. Pre-incubation of the toxin mixture with recombinant antibodies led to a 50% attenuation of this effect. This finding reflected well the poor TcdA reactivity of the three antibody types analysed.

In order to elicit cytotoxic effects, TcdA and TcdB must be internalised into the host cell via endocytosis. Receptor binding is the first essential step in the process of cell entry. In the case of TcdA, a disaccharide harbouring a Galβ1-4GlcNac motif has been identified as a toxin binding structure. This disaccharide is found on the I, X and Y blood antigens present in a variety of cells, and these antigens have been shown to act as receptors for TcdA [39]. The presence of these (and/or other) TcdA receptors on canine epithelial cells remains to be determined. In addition, whilst TcdA has been shown to induce disruption of tight junctions, fluid influx, diarrhoea, inflammation and neutrophil recruitment in human patients [18], several aspects of TcdA and TcdB cytotoxicity in canines are still underexplored.

This motivated us to analyse the toxins’ effects on the barrier integrity in three-dimensional small and large intestinal floating basal-out and apical-out organoids. This approach allowed controlled antibody/toxin delivery and subsequent study of toxin-induced damage without needing to perform laborious microinjections [40,41]. Given that all three antibody types did not sufficiently react with TcdA in ODM culture, and that caprine sIgA failed to neutralise TcdB, 3D experiments were conducted with both toxins, but neutralisation assays exclusively focussed on the TcdB reactivity of recombinant anti-TcdB antibodies.

TcdA reduced barrier integrity in small and large intestinal organoids, irrespective of the intestinal cells’ orientation. Since the discovery of I, X and Y blood antigens as TcdA receptors in 2005, more recent evidence points to the existence of several alternative pathways mediating TcdA entry into cells. Various domains of TcdA have been found to interact with numerous cell surface molecules to hijack internalisation into endosomes [42,43].

TcdB only had a cytotoxic effect on basal-out organoids representing the small intestine. Neither small nor large intestinal apical-out organoids appeared to be affected by TcdB. This finding is suggestive of TcdB receptors being predominantly, or even exclusively, expressed on the basolateral surface of cells, which is in accordance to previous literature showing that TcdB mainly effects the basolateral cell-surface [44]. This also indicates that TcdB receptor expression may significantly differ between canine small and large intestine.

Compared to previous studies that used only monolayers to test *C. difficile* toxins [45,46], we have established a system comparing 2D and 3D effects. This clearly shows that 2D experiments cannot be extrapolated well into 3D in vitro settings, as they might be substantially different. Of course, this also holds true to extrapolating data to mouse models or even human treatment, which, in part, explains why the majority of newly developed drugs fail on their way to clinical use. Investigating different species and in vitro setups will aid in the development of new treatment approaches, especially for diseases relevant to a One Health context, where every affected species might react differently to a pathogen and/or the respective treatment.

Studies in animal models and human patients have substantially advanced our understanding of TcdA and TcdB pathogenicity, but also revealed considerable differences between species regarding intestinal vulnerability to these toxins. For example, TcdB is not enterotoxic in rabbit ileal and colonic loops, or hamster and mouse ileal loops, but causes severe jejunal lesions in these species. On the other hand, ex vivo studies conducted in human colonic explants showed that TcdB induces pathobiological changes consistent with enterotoxicity [47]. Of the three previously determined receptors for TcdB (CSPG4, Fzd1,2,7 and PVRL3/Nectin3) [48,49,50,51], CSPG4 was found to be absent in mouse colonic epithelial cells [49,52]. Whether a similar absence of particular receptors could be responsible for the finding that canine colonic organoids are less susceptible to TcdB damaging their barrier integrity remains to be determined in future studies.

Species-specific nutrition and environmental conditions make highly divergent demands on the respective gastrointestinal system. This likely explains interspecies variations with respect to resistance/sensitivity of bowel segments to TcdA and TcdB and emphasizes that enterotoxic activities of TcdA and TcdB observed in animal models should be confirmed in the target species to reach final conclusions. In this context, canine intestinal organoids represent a particularly interesting research tool: As “micro copies” of the organ they represent and from which they originate, they allow the recapitulation of many important physiological and pathobiological mechanisms in dogs. In addition, close resemblances between healthy and disease-affected canines and the human gut have been demonstrated [26]. Consequently, canine intestinal organoids can also serve as model for human intestine, thus helping overcome the limitations of rodent systems.

However, one should cautiously evaluate toxin effects in different species instead of extrapolating results from one species to another. Human organoid and/or intestinal epithelial systems have been used previously to assess the effects of TcdA and TcdB [45,46,53]. Using 2D and 3D approaches on canine intestinal organoids, we showed that toxin effects are not the same in small and large intestinal-derived organoids, even from the same species, so results from human studies may not be transferred directly to other organisms. In addition, our results provided proof of concept that monoclonal sIgA molecules are equally effective as existing IgG antibodies as a more stable treatment option.

## 4. Materials and Methods

### 4.1. Recombinant Monoclonal sIgA

Monoclonal recombinant anti-TcdB mIgA2 antibodies and the human SC component (hSC) were produced in recombinant Chinese Hamster Ovary cells (CHO-K1), as described by Bhaskara et al., 2021 [31]. In brief, recombinant anti-TcdB-mIgA2-expressing and hSC-expressing CHO-K1 cells were cultured in a humidified shaking incubator at 37 °C/5% CO_2_/160 rpm in 2 L fed batch cultures, each for 10 days in HyClone ActiPro medium (ThermoFisher Scientific, Vienna, Austria). The cultures were fed on days 3, 5, 6 and 7 with 4% (*v*/*v*) of HyClone Cell Boost 7a and 0.4% (*v*/*v*) of HyClone Cell Boost 7b (ThermoFisher Scientific, Vienna, Austria). On day 10, cell culture supernatants were harvested by centrifugation for 20 min at 1200 rpm, mixed with Sartoclear Dynamics Lab V Kit diatomaceous earth (Sartorius, Vienna, Austria) at 200 g/L and filtered through sterile 0.22 μm filters. Anti-TcdB-sIgA and hSC supernatants were mixed at a 2:1 volume ratio and the mixture was incubated, shaking overnight at room temperature to potentiate the formation of hSC-sIgA complexes.

Then, the antibody-hSC mixture was diluted 1:2 with 20 mM Tris pH 8.0 to reach a conductivity of 8 mSi/cm and loaded onto an Äkta anion exchange column/20 mM Tris pH 8.0 (Cytiva Life Sciences, Vienna, Austria) at 2.5 mL/min. The column was washed with 20 mM Tris pH 8.0 and the samples were eluted with a salt gradient of 0–30% high salt buffer (20 mM Tris/1 M NaCl). Fractionated samples were loaded on SDS PAGE gels (NuPAGE 4–12% Bis-Tris Midi Gel; ThermoFisher Scientific, Vienna, Austria), and stained with InstantBlue (Expedeon, VWR, Vienna, Austria). Then fractions containing sIgA but not free dimeric (dIgA) molecules were pooled, concentrated using Amicon Ultra-15 Centrifugal Filter Units (Merck-Millipore, Darmstadt, Germany) and loaded onto a Sephadex^®^ G-200 size exclusion column in 0.1 M Borate buffer pH 8.0 (Sigma-Aldrich, Vienna, Austria). Following washing with this buffer, protein fractions were eluted with 0.1 M Glycine pH 2.7 into Tris buffer pH 9.0. Finally, protein aliquots were loaded on SDS-PAGE gels. Fractions containing pure sIgA were pooled and filtered through sterile 0.22 μm filters. Borate buffer was substituted by phosphate-buffered saline (PBS), sIgA was concentrated on Amicon Ultra-15 Centrifugal Filter Units to 1.7 mg/mL and kept at 4 °C until use.

### 4.2. Isolation of Caprine Whey sIgA

Whey was collected from a local dairy farm and pooled from several goats. Then, 1 L of pooled caprine whey was centrifuged at 10,000× *g* for 1 h to separate large particles. The obtained supernatant was filtered using a Sartoban P0.2 µm depth filter (Sartorius, Vienna, Austria) and a peristaltic pump. Filtration comprised a 0.45 µm pre-filtration and a 0.2 µm filtration step. Using an Äkta flux system (Cytiva Life Sciences, Vienna, Austria), the solution was subsequently concentrated using 2 × 100 kDa membranes with a total filter size of 0.01 m^2^. The initial flow rate of 150 mL/min was steadily decreased to keep transmembrane pressure (TMP) below 1.5 bar. Using this approach, 950 mL of starting volume was concentrated by a factor of 29.7 to a final volume of 32 mL, which was subjected to size exclusion chromatography. Finally, preparative chromatography was carried out using the Äkta pure system (Cytiva Life Sciences, Vienna, Austria) with a Superdex 200 26/60 preparative grade column (CV = 320 mL), allowing separation of 10 to 600 kDa proteins. PBS was used as an equilibration and elution buffer for isocratic elution. Ten millilitres of the concentrated sample (9.4% of CV) per run was transferred to a ten millilitres super loop and elution was carried out for a total of one point two millilitres CV. Quantification of sIgA concentrations from isolated goat whey fractions was carried out by competitive ELISA (Antibodies-online, Aachen, Germany).

### 4.3. ELISA

The binding potential of antibodies to TcdA and TcdB (BioTrend, Vienna, Austria) was assessed by indirect ELISA. The latter was conducted for total volumes of 100 µL/well in high-binding microtitre plates (Greiner bio-one, Kremsmünster, Austria). All washing steps were carried out using PBS-T (PBS with 0.05% Tween 20; Sigma-Aldrich, Vienna, Austria). Plates were coated with TcdA or TcdB at a concentration of 1 µg/mL in PBS and incubated overnight at 4 °C without shaking and then washed three times with PBS-T. After a blocking step of 20 min with PBS-T, the three goat whey fractions showing the purest sIgA profile (5B3, 5B4, 5B5) and control antibodies bezlotoxumab and recombinant sIgA diluted in PBS containing 1% bovine serum albumin (BSA, Carl Roth, Karlsruhe, Germany) were incubated by shaking at 40 rpm for one hour at room temperature. Concentrations tested were 5 µg/mL, 10 µg/mL, 20 µg/mL, 40 µg/mL, 80 µg/mL and 160 µg/mL for caprine whey sIgA and 5 µg/mL for controls, respectively. Unbound proteins were removed by washing three times. Horseradish peroxidase-conjugated secondary antibodies against goat IgA or human IgG (both BioRad, Vienna, Austria) were diluted 1:10,000 in PBS containing 1% BSA, added to the appropriate wells, and incubated for 45 min at room temperature, shaking at 40 rpm. After washing, substrate (3,3′,5,5′-Tetramethylbenzidine (TMB, Sigma-Aldrich, Vienna, Austria)) was added. Following signal development, the reaction was stopped with 2 N HCl (Carl Roth, Karlsruhe, Germany). The absorbance was measured at 450 nm on a TECAN plate reader (Tecan Life Sciences, Männedorf, Switzerland). Analysis was performed from two independent experiments, each carried out in duplicates.

### 4.4. Organoid Culture

Canine intestinal crypts were isolated from jejunum and colon according to Kramer et al., 2020 [28]. Tissue sampling was approved by the institutional ethics committee, in accordance with Good Scientific Practice guidelines and Austrian legislation. Based on the guidelines of the institutional ethics committee, the use of tissue material collected during therapeutic excision or post-mortem is included in the University’s “owner’s consent for treatment”, which was signed by all patient owners. The growth medium consisted of 37% basal medium (Advanced DMEM/F12 supplemented with 2 mM GlutaMAX and 10 mM HEPES), 1 × B27 (Invitrogen, ThermoFisher Scientific, Vienna, Austria), 1 mM N-acetylcysteine, 10 nM Gastrin (Sigma-Aldrich, Vienna, Austria), 100 ng/mL Noggin, 500 nM A8301, 50 ng/mL HGF, 100 ng/mL IGF1, 50 ng/mL FGF2 (PeproTech, Rocky Hill, NJ, USA), 10% (*v*/*v*) Rspondin1 and 50% (*v*/*v*) Wnt3a conditioned media. For the first two days of culture, 50 ng/mL mEGF (ThermoFisher Scientific, Vienna, Austria) and 10 µM Rock-inhibitor Y-27632 (Selleck Chemicals, Houston, TX, USA) were added. The growth medium was changed every two to three days. Weekly passaging at 1:4 to 1:8 split ratios was achieved by mechanical disruption using flame-polished Pasteur pipettes. Brightfield images were acquired using a DMi8 microscope (Leica Camera AG, Wetzlar, Germany).

### 4.5. Generation of Organoid-Derived Monolayers

To analyse the cytotoxic effects of *C. difficile* toxins A and B, organoid-derived monolayers were established. To this aim, organoids were released from the Geltrex matrix via repeated pipetting. Organoids were then trypsinised, and single cells were counted. Subsequently, 15,000 cells/well were seeded in 96-well plates pre-coated with 100 µg/mL Geltrex (ThermoFisher Scientific, Vienna, Austria) diluted in a basal medium at 37 °C for 1 h.

### 4.6. Sulforhodamine B Cytotoxicity Assay

When almost reaching confluence, cells were treated with 7.5 ng/mL TcdA, 100 ng/mL TcdB, or a combination of both. The applied toxin concentrations were predetermined as the respective IC50. The neutralisation assay was carried out by pre-incubating TcdA and TcdB with 50 µg/mL monoclonal antibody bezlotoxumab (Merck, Darmstadt, Germany), monoclonal recombinant sIgA, or sIgA isolated from goat whey for 2 h at 37 °C prior to cell treatment. Used concentrations equalled to a 14,000-fold and 930-fold molar excess of bezlotoxumab and a 5300-fold and 350-fold molar excess of sIgA to TcdA and TcdB, respectively. After 24 h, cells were fixed and subjected to sulforhodamine B (SRB) cytotoxicity assays [54]. In brief, cells were fixed with 10% trichloroacetic acid (TCA) for one hour at 4 °C and then rinsed with tap water four times. After air-drying, cells were stained with 0.057% SRB in 1% acetic acid for 30 min. Subsequently, cells were rinsed with 1% acetic acid four times to remove excess SRB before air-drying again. Cell-bound dye was then solubilised in 10 mM TRIS and extinction was measured at 488 nm using a fluoro spectrometer (Glomax^®^ Explorer, Promega, Vienna, Austria).

### 4.7. Generation of Floating Basal-Out and Apical-Out Organoids

Apical-out and floating basal-out organoids were generated as described previously [55]. Organoids were harvested using Cultrex^®^ Organoid Harvesting Solution (Bio-Techne, Minneapolis, MN, USA) for 1.5 h at 4 °C, under constant shaking. Thereafter, organoids were washed twice with basal medium, resuspended in growth medium and seeded in multiwell plates treated with Anti-Adherence Rinsing Solution (Stemcell Technologies, Vancouver, Canada) to prevent organoid attachment to the surface. To generate floating basal-out organoids, 7.5% Geltrex (ThermoFisher Scientific, Waltham, MA, USA) was added to the culture medium. Organoids were incubated for 72 h in a humidified incubator with 5% CO_2_ prior to further use.

### 4.8. DAPI/Phalloidin Staining

Organoids were fixed with 2% paraformaldehyde (PFA) and stained with Phalloidin (2.5 µg/mL in PBS; Alexa Fluor 546, Invitrogen, ThermoFisher Scientific, Vienna, Austria) to visualise actin filaments, and with 4′,6-diamino-2-phenylindole (DAPI; 4 µg/mL in PBS; Sigma-Aldrich, Vienna, Austria) nuclear staining for 45 min. Images were acquired using a Zeiss LSM880 confocal microscope (Zeiss, Jena, Germany).

### 4.9. FITC-Dextran Barrier Integrity Assay

Basal-out and apical-out organoids were embedded in 18-well slides with glass bottoms (Ibidi, Gräfelfing, Germany) in Geltrex and treated with 2.5 ng/mL TcdA, 7.5 ng/mL TcdB, or a combination of both with and without neutralising antibodies. This treatment was achieved by replacing the entire medium covering the Geltrex dome with fresh toxins diluted in fresh medium. As in SRB assays, the toxin:antibody:medium mixture was pre-incubated for 2 h at 37 °C prior to cell treatment. Toxin:antibody ratios corresponded to those in the SRB assays, but the concentration of both toxins was reduced to induce damage to the epithelial barrier whilst keeping the organoids alive due to the apparently higher sensitivity of whole organoids compared to ODMs. After 23 h, FITC-dextran 4000 (Sigma-Aldrich, Vienna, Austria) was added to the cultures at a final concentration of 1 mM and images were acquired using a Zeiss LSM880 confocal microscope after 24 h of incubation, as described in Bardenbacher et al., 2020 [56]. Images were analysed using Fiji ImageJ [57] for the calculation of grey values within each organoid.

### 4.10. Statistics

All statistical analyses were performed using GraphPad Prism Version 9.3.0.

## Figures and Tables

**Figure 1 ijms-24-03867-f001:**
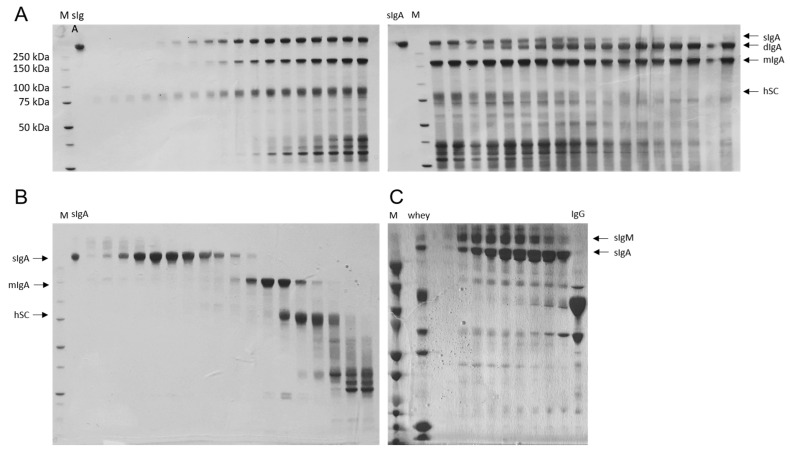
SDS-PAGE of different immunoglobulin-fractions: (**A**) SDS-PAGE analysis of sIgA fractions obtained by anion exchange chromatography. sIgA molecules were eluted under lower salt concentration conditions than dIgA from the AIEx column, allowing their separation. From left to right: subsequently obtained AIEx fractions. (**B**) SDS-PAGE analysis of sIgA obtained by size exclusion chromatography purification. sIgA molecules were efficiently separated by size from mIgA, hSC and other protein contaminants. Fractions were loaded from left to right. (**C**) SDS-PAGE analysis of sIgA-containing fractions extracted from goat whey by chromatography. Fractions were loaded from left to right. M (protein weight marker): BioRad Precision Plus Protein Unstained (A,B), BioRad Precision Plus Protein Dual Color Standard (**C**).

**Figure 2 ijms-24-03867-f002:**
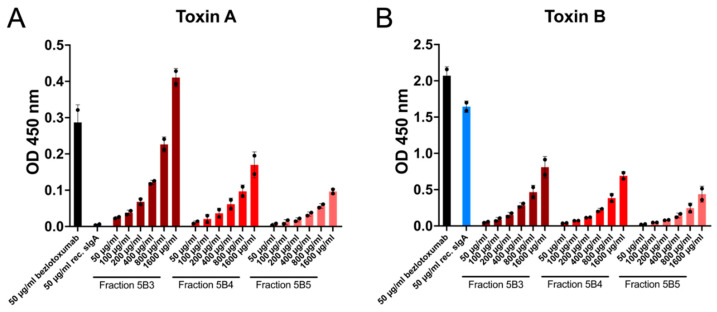
Neutralising potential of different antibodies against *C. difficile* toxin A (TcdA) and toxin B (TcdB). (**A**) All antibodies showed very low binding affinity to TcdA, with caprine whey sIgA fractions slightly increasing at higher concentrations. (**B**) Caprine whey sIgA fractions had a much lower binding potential against TcdB than bezlotoxumab and monoclonal recombinant sIgA. Data are represented as mean ± standard deviation (n = 2).

**Figure 3 ijms-24-03867-f003:**
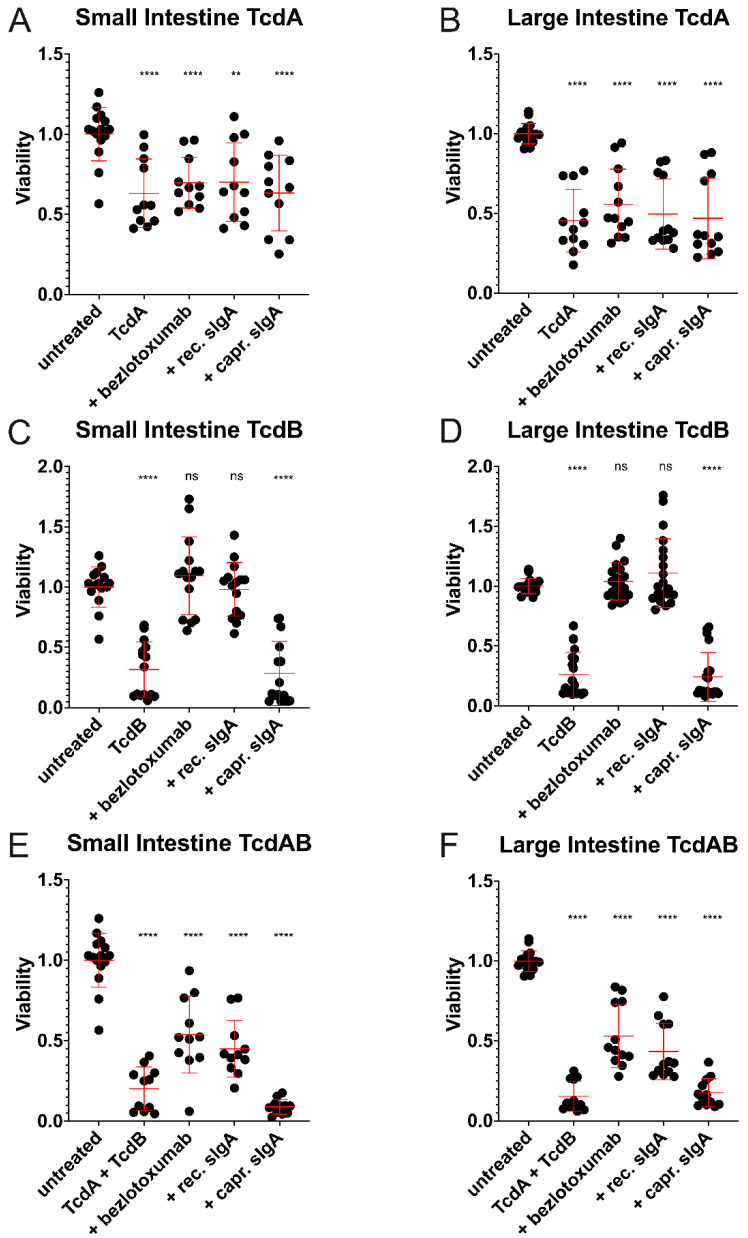
Respective potential of bezlotoxumab, recombinant sIgA and caprine sIgA to neutralise TcdA and/or TcdB, as revealed by sulforhodamine B cytotoxicity assays in ODM culture. Viability of ODMs from canine small and large intestinal organoids treated with TcdA (**A**,**B**), TcdB (**C**,**D**) or a combination of both toxins (**E**,**F**) following their incubation with bezlotoxumab, recombinant sIgA or caprine sIgA is shown. Student’s *t*-tests were performed with the viability of untreated control ODMs serving as a reference. Error bars indicate standard deviation from arithmetic means. ** *p* < 0.01, **** *p* < 0.0001, ns = non-significant.

**Figure 4 ijms-24-03867-f004:**
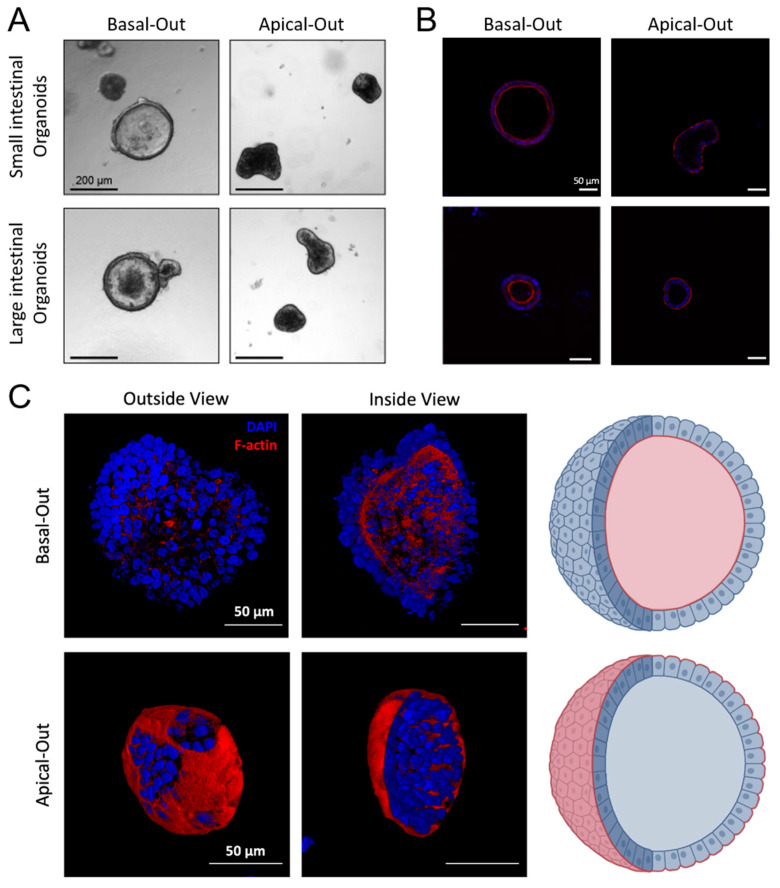
Characterisation of canine small and large intestinal apical out organoids. (**A**) Basal-out and apical-out organoids showed different morphologies as documented by brightfield microscopy. (**B**) Basal-out and apical-out organoids were stained with DAPI (blue) and Phalloidin (red) to visualise F-actin-rich microvilli on the apical cell surface of intestinal epithelial cells. (**C**) Three-dimensional rendering of basal-out and apical-out organoids, including a scheme depicting dense microvilli on the inside in basal-out organoids and on the outside in apical-out organoids.

**Figure 5 ijms-24-03867-f005:**
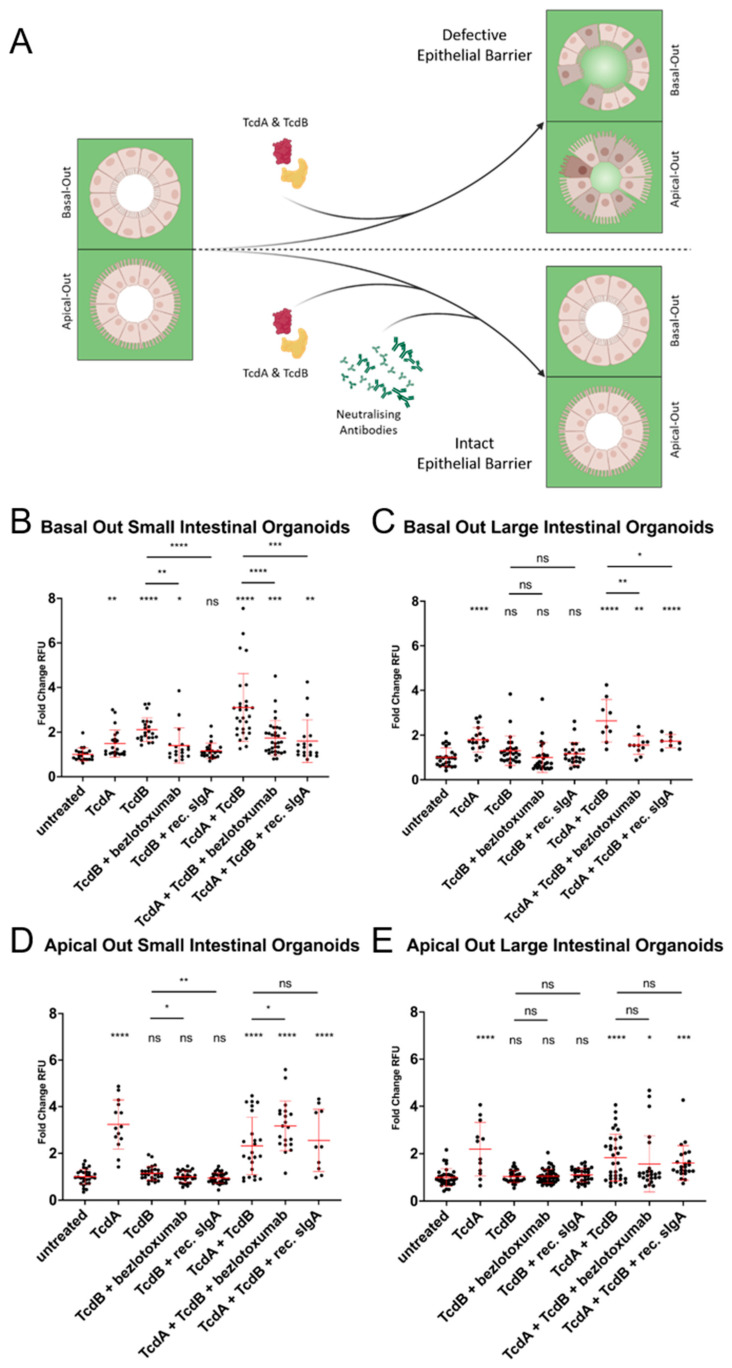
Different effects of TcdA and TcdB on basal-out and apical-out organoids in FITC-dextran barrier integrity assays. (**A**) A scheme representing the experimental approach. Small and large intestinal basal-out (**B**,**C**) and apical-out organoids (**D**,**E**) treated with TcdA, TcdB or a combination of both toxins. TcdB and the combination of TcdA and TcdB were pre-incubated with two different antibodies (bezlotoxumab and recombinant sIgA). Student’s *t*-tests were performed using untreated controls as a reference, if not indicated otherwise. Error bars indicate standard deviations from the arithmetic means. RFU = relative fluorescent units, * *p* < 0.05, ** *p* < 0.01, *** *p* < 0.001, **** *p* < 0.0001, ns = non-significant.

## Data Availability

All the data generated or analyzed during this study are included in this article. Raw data are available on request from the corresponding author upon request.

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
