# Peer review of "Neutralising Effects of Different Antibodies on *Clostridioides difficile* Toxins TcdA and TcdB in a Translational Approach"

_ijms, 2023, doi:10.3390/ijms24043867_

Round 1
Reviewer 1 Report
The current article entitled “Neutralising effects of different antibodies on Clostridioides difficile toxins TcdA and TcdB in a translational approach” by Csukovich et al. used canine organoid model system to assess the effects of toxins and their neutralizing antibodies. They found that recombinant, but not natural antibodies effectively neutralizes C. difficile toxins in canine intestinal organoid cultures in ex-vivo conditions.
Comments:
Major:
1. Although authors performed viability on monolayer epithelial cells derived from the intestinal organoids and showed in combination with toxins and recombinant antibodies neutralizes the effects. The morphological characterization should be performed by assessing the area of the intestinal organoids in presence of toxins and in combination with toxins and antibodies. This will help in assessing effects in real 3 dimensional organoid settings.
2. As authors showed compromised intestinal barrier in presence of toxins TcdA while this effect is neutralized in presence of antibodies bezlotoxumab and recombinant sIgA. However, there is complete lack of knowledge about the mechanistic aspects. It would be interesting to perform at least RT-qPCR analysis and/or western blot/immunofluorescence for the genes/proteins involved in maintaining tight junctions in the intestinal organoids such as cluadins, zonula occludins, MLCKs etc. before and after treatments with toxins and these neutralizing antibodies.
Minor
1. In the results sections references for figure citations are missing “error”
2. Figure 2 legends- indicates replicates, and statistics, p value
Author Response
The current article entitled “Neutralising effects of different antibodies on Clostridioides difficile toxins TcdA and TcdB in a translational approach” by Csukovich et al. used canine organoid model system to assess the effects of toxins and their neutralizing antibodies. They found that recombinant, but not natural antibodies effectively neutralizes C. difficile toxins in canine intestinal organoid cultures in ex-vivo conditions.
We thank the reviewer very much for the insightful comments that show that our manuscript clearly reflects our study. The comments are very relevant to our future research plans and ongoing projects that deal with many of these ideas following the successful set-up of our model system to efficiently test for barrier disrupting activities of different toxins and their effects on cell survival. The comments are highly appreciated and we hope we answered all open questions to the reviewer’s entire satisfaction (see below).
Comments:
Major:
- Although authors performed viability on monolayer epithelial cells derived from the intestinal organoids and showed in combination with toxins and recombinant antibodies neutralizes the effects. The morphological characterization should be performed by assessing the area of the intestinal organoids in presence of toxins and in combination with toxins and antibodies. This will help in assessing effects in real 3 dimensional organoid settings.
There are two sides to this comment:
- We did analyse viability with a sulforhodamine B assay to have a more final output of survival rather than analysing cell rounding as other authors did when analysing the effects of difficile toxins. If you analyse cell rounding via microscopy, these cells can either end up dead or recover and “re-integrate” into the monolayer/epithelium. Therefore, a morphological analysis of monolayers does not seem to give us any new insights and is highly subjective depending on the area photographed.
- Analysing the morphological changes in terms of area in entire three-dimensional organoids is not really feasible. If one would analyse the area from the confocal images of the FITC-dextran barrier integrity assays, there is the problem that organoids are usually not perfectly sphere-shaped, may present budding structures etc. and therefore the compromise of choosing the mid-layer of each organoid for this kind of analysis would probably skew these results. On the other side, one could use bright field images. However, here the problem of unclear edges of the organoids exists.
Therefore we think that our 3D analysis is the best option to analyse toxin effects on organoid/barrier integrity available to judge the effects as objectively as possible without relying too much on perfectly sphere-shaped organoids.
- As authors showed compromised intestinal barrier in presence of toxins TcdA while this effect is neutralized in presence of antibodies bezlotoxumab and recombinant sIgA. However, there is complete lack of knowledge about the mechanistic aspects. It would be interesting to perform at least RT-qPCR analysis and/or western blot/immunofluorescence for the genes/proteins involved in maintaining tight junctions in the intestinal organoids such as claudins, zonula occludins, MLCKs etc. before and after treatments with toxins and these neutralizing antibodies.
We point out in the discussion that indeed several aspects of C. difficile toxin mechanisms are unknown, especially in the canine system. This is definitely a very important point for future studies and TJ protein expression should definitely be analysed in more depth in the near future.
However, finding antibodies that actually work reliably in the canine system for detection in Western blot or IF experiments is currently ongoing but appears to be a very tough job as many antibodies that bind human or mouse proteins do not react to canine samples, even though sequence similarity is at a full 100 %. Therefore, this requires more research together with expression analysis of possible toxin receptors and intracellular changes following toxin binding/internalisation that will be part of a separate study. In the presented study, we have focused on addressing the usability of our model system and three antibodies for future applications. We are very sorry we are not able to include these analyses as part of the manuscript as of now, all the more since FITC-dextran would most likely interfere with relevant RT-PCR experiments. Consequently, experiments would have to be repeated to generate organoids not incubated with FITC-dextran.
Minor
- In the results sections references for figure citations are missing “error”
We are very sorry to hear this as this has happened during earlier submissions to MDPI as well. These links seem to go missing somehow when different people using different versions of MS Word open the manuscript. As this was not a problem in the version provided by MDPI for revision, we have not made any adaptations here, but it will be corrected in case the problem should arise again.
- Figure 2 legends- indicates replicates, and statistics, p value
We added the information “n = 2” to the figure legend.
Statistics and p-values would not be meaningful given the fact we only repeated this twice to get an idea of which whey fraction was most efficiently binding the toxins.
Reviewer 2 Report
In this manuscript, the authors tried to identify the neutralizing effects of recombinant and natural antibodies on clostridioides difficile toxins A and B in vitro system. They established canine organoid-derived monolayers and 3D basal-out and apical-out organoids to address the questions and revealed that recombinant, but not natural antibodies effectively neutralized clostridioides difficile toxins. The experiments and results are solid. However, some questions need to be addressed before publishing.
Comments:
1. Why do the authors not use human organoids to replace canine organoids?
2. Could the authors explain the difference between basal-out and apical-out organoids and the uptake of TcdA and TcdB?
Author Response
In this manuscript, the authors tried to identify the neutralizing effects of recombinant and natural antibodies on Clostridioides difficile toxins A and B in vitro system. They established canine organoid-derived monolayers and 3D basal-out and apical-out organoids to address the questions and revealed that recombinant, but not natural antibodies effectively neutralized Clostridioides difficile toxins. The experiments and results are solid. However, some questions need to be addressed before publishing.
We thank the reviewer for the review of our manuscript and hope that our answers provided below are satisfactory:
Comments:
- Why do the authors not use human organoids to replace canine organoids?
This is due to our research focus on animals, especially pet animals, as members of the Unit for Small Animal Internal Medicine at the University of Veterinary Medicine, Vienna. We could have only included human organoids as reference, but this would have led to much higher costs and possibly not entirely comparable results as different media would have had to be used for organoid culture. Additionally, we cannot guarantee the same degree of differentiation in canine and human organoids at the same time, further increasing complexity.
- Could the authors explain the difference between basal-out and apical-out organoids and the uptake of TcdA and TcdB?
Lines 406-472 of our discussion deal with the toxins’ effects. While previous studies suggest that TcdA can bind virtually everywhere on the cell surface to get internalized, TcdB can only bind to receptors on the basolateral surface of the cell. Which receptors TcdB binds to is not yet fully understood. However, our results are in accordance to previous literature indicating that only basal-out organoids seem to be negatively affected by TcdB. As highlighted in our discussion, a lot of research is still necessary to elucidate the exact mechanisms of TcdA and TcdB effects on the intestinal epithelium – especially in the canine system, as this is the first study to use canine intestinal cells. We believe that our newly established model will be extremely valuable for future research in the field.